# Layer-by-Layer Electrode Fabrication for Improved Performance of Porous Polyimide-Based Supercapacitors

**DOI:** 10.3390/ma15010004

**Published:** 2021-12-21

**Authors:** Niranjala Fernando, Hugo Veldhuizen, Atsushi Nagai, Sybrand van der Zwaag, Amor Abdelkader

**Affiliations:** 1Department of Engineering, Talbot Campus, Bournemouth University, Fern Barrow, Poole BH12 5BB, UK; nweerahannadige@bournemouth.ac.uk; 2Faculty of Aerospace Engineering, Delft University of Technology, Kluyverweg 1, 2629 Delft, The Netherlands; h.v.veldhuizen@tudelft.nl (H.V.); s.vanderzwaag-1@tudelft.nl (S.v.d.Z.); 3Department of Electrical and Electronic Information Engineering, Toyohashi University of Technology, Hibarigaoka-1-1 Tenpakucho, Toyohashi 441-8580, Japan; nagai.atsushi.rn@tut.jp

**Keywords:** nanoporous polymer, supercapacitor, polyimide, electrochemistry, electrode fabrication, layer-by-layer

## Abstract

Nanoporous polymers are becoming increasingly interesting materials for electrochemical applications, as their large surface areas with redox-active sites allow efficient adsorption and diffusion of ions. However, their limited electrical conductivity remains a major obstacle in practical applications. The conventional approach that alleviates this problem is the hybridisation of the polymer with carbon-based additives, but this directly prevents the utilisation of the maximum capacity of the polymers. Here, we report a layer-by-layer fabrication technique where we separated the active (porous polymer, top) layer and the conductive (carbon, bottom) layer and used these “layered” electrodes in a supercapacitor (SC). Through this approach, direct contact with the electrolyte and polymer material is greatly enhanced. With extensive electrochemical characterisation techniques, we show that the layered electrodes allowed a significant contribution of fast faradic surface reactions to the overall capacitance. The electrochemical performance of the layered-electrode SC outperformed other reported porous polymer-based devices with a specific gravimetric capacitance of 388 F·g^−1^ and an outstanding energy density of 65 Wh·kg^−1^ at a current density of 0.4 A·g^−1^. The device also showed outstanding cyclability with 90% of capacitance retention after 5000 cycles at 1.6 A·g^−1^, comparable to the reported porous polymer-based SCs. Thus, the introduction of a layered electrode structure would pave the way for more effective utilisation of porous organic polymers in future energy storage/harvesting and sensing devices by exploiting their nanoporous architecture and limiting the negative effects of the carbon/binder matrix.

## 1. Introduction

The development of low cost, eco-friendly and high-performance energy storage devices is becoming crucial to meeting modern energy storage demand in various applications, including electric vehicles, portable electronics, and power grids. Among them, supercapacitors (SCs) gained significant attention as a promising energy storage device due to their outstanding energy density, power density and long cycle life [1,2]. There are two types of supercapacitor electrodes; one that uses the electric double layer phenomenon and one based on fast faradic surface reactions (pseudocapacitive SCs). High-performance electrode materials should have properties such as an electrical conductivity, a large specific surface area, and good wettability by the electrolyte in both cases. Several materials have been used to fabricate electrodes for supercapacitors, including carbon nanotubes [1,3], graphene [4,5], metal oxides nanoparticles [6,7,8], transition metal dichalcogenides [9], and carbon-based composites [10,11].

Next to these materials are conductive polymers, which have become increasingly popular alternatives as novel active materials for SCs [12,13,14,15]. Polymers as active materials for supercapacitors have several advantages over inorganic active materials, such as high theoretical specific capacitance, low cost and ease of large-scale production [16]. Furthermore, small molecular changes in the originating monomeric building blocks can yield a library of polymers with macroscopically different properties, meaning that polymers are highly tuneable. In the field of SCs, this tunability is reflected in the fact that redox-active segments can be implemented into the polymer-backbone, providing an additional pseudocapacitive energy storage feature. However, conductive polymers still experience limitations, such as poor cycle life and rate kinetics due to repeated swelling and shrinking of the polymer structure during the charge–discharge processes. Additionally, polymer particle aggregation makes it challenging to engineer electrodes that maintain the high theoretical specific capacitance [17].

Alternatively, nanoporous organic polymers emerged as an attractive replacement for linear conductive polymers. Their porous architecture circumvents problems such as low surface areas and aggregation of linear polymers. Thus, nanoporous polymers, in particular, allow the efficient diffusion of ions over an enormous surface area of active material, which significantly improves the capacity [18,19]. However, nanoporous polymers introduce the problem of not providing efficient conductive pathways. To alleviate the poor conductivity of these materials, the concept of hybridising them with conductive materials (often carbon-based) has been suggested [20,21,22]. However, the addition of conductive materials, even when it contributes to the capacitance, reduces the specific capacity of the electrodes below the theoretical value of the nanoporous polymers. Moreover, complex techniques are often required to ensure good adhesion between the conductive agent and redox-active polymer, which hinders their practical applications. Therefore, it is desirable to engineer electrodes that can utilise the high surface area of the porous polymers without sacrificing the specific capacitance.

In the study presented here, we utilised a layer-by-layer electrode fabrication approach to produce bilayer conductive carbon/perylene diimide-based porous polymer electrodes as an alternative to the traditional casting of a pre-mixed paste of the active materials, binder and conductive agent (Figure 1). This electrode preparation approach allows us to prepare electrodes with maximum utilisation of the porous polymer as the active supercapacitive material by enhancing the exposure of active materials to the electrolyte, as has also been exploited by other research groups in the field of 1D polymers [23]. The formation of a layer-by-layer assembly can be explained through intermolecular interactions and is dependent on the chemical nature of the separate layers.

Polyimide porous polymers have been extensively used in gas storage and separation applications [24,25,26,27], but the investigation of their potential in the field of energy storage has been limited due to their poor conductivity [28,29,30,31]. However, the redox-active segments within their polymer backbone make them interesting to study, especially regarding energy storage applications. One-dimensional perylene diimide-based materials have been widely exploited in the field of photovoltaics and batteries. Often, high electron mobilities in such materials are observed as an effect of their favourable molecular π–π interactions [32]. In addition, these aromatic polyimides are redox-active and their carbonyl groups can participate in reversible lithium complexation [33]. The combination of both of these properties, integrated within a porous framework, formed the idea that a porous perylene diimide-based polymer may exhibit good electrochemical performance in the application presented here.

## 2. Experimental

### 2.1. Materials

All chemicals were commercially available and used without further treatment. Chemicals including; 3,4,9,10-perylenetetracarboxylic dianhydride (PTCDA); 1,3,5-tris(4-aminophenyl)benzene (≥93%, TAPB); isoquinoline; imidazole; *meta*-cresol and dimethyl carbonate were all purchased from TCI Europe N.V., Zwijndrecht, Belgium. In addition, 1-methyl-2-pyrrolidinone (NMP), poly(vinylidene fluoride) (PVDF) and 1.0 M LiPF6 in ethylene carbonate/ethyl methyl carbonate = 1/1 (*v*/*v*), battery grade were purchased from Sigma Aldrich, Gillingham, UK.

### 2.2. Synthesis of **Per-TAPB-PPI**

Figure 1 shows the chemical structures of TAPB (204 mg, 0.58 mmol) and PTCDA (345 mg, 0.88 mmol), which were weighed and transferred to a 25 mL ampoule. An amount of 10 mL of a 1:1 (*v*/*v*) mixture of *meta*-cresol and imidazole was added to the ampoule, and the mixture was sonicated for five minutes. Then, 0.1 mL of isoquinoline was added, after which the ampoule was degassed via three freeze–pump–thaw cycles. Finally, the ampoule was flame-sealed, allowed to reach room temperature, and placed in an oven at 200 °C for 5 days. After polymerisation, the precipitate was washed with methanol (2 × 30 mL) and acetone (2 × 30 mL). The solid was dried in a vacuum oven (60 °C) and subsequently washed in a Soxhlet extractor using THF for 16 h. The resulting polymer, **Per-TAPB-PPI**, was finally dried overnight in a vacuum oven at 60 °C and obtained as a dark red powder with a yield of 91%.

### 2.3. Molecular and Microstructural Characterisation

FT-IR spectra were recorded on a PerkinElmer Spectrum 100 FT-IR Spectrometer (Perkin Elmer, Bruxelles, Belgium) with the Universal ATR Accessory over a range of 4000 to 650 cm^−1^. TGA analyses were performed from room temperature to 860 °C, under nitrogen atmosphere at a heating rate of 10 °C/min using a Perkin Elmer TGA 4000 (Perkin Elmer, Bruxelles, Belgium). Before the measurement, the samples were dried at 130 °C for one hour under a nitrogen atmosphere. The nitrogen sorption isotherms (at 77 K) were measured on a Tristar II 3020 Micromeritics instrument (Micromeritics B.V., Eindhoven, The Netherlands). The porous polymer was degassed before the measurement at 130 °C under vacuum for 16 h. Quenched-Solid Density Functional Theory (QSDFT) simulations based on the experimental nitrogen adsorption isotherms were performed to calculate the pore size distributions. The chosen model was the carbon, adsorption kernel, slit/cylindrical/spherical pore model, a standardised model in the VersaWin Gas Sorption Data Reduction Software (Quantachrome Instruments, Boynton Beach, FL, USA). Scanning electron microscopy (SEM) images were recorded with a JEOL JSM-840 SEM (JEOL Europe B.V., Nieuw-Vennep, The Netherlands): materials were deposited onto a sticky carbon surface on a flat sample holder, vacuum-degassed, and subjected to gold sputtering. PXRD data were collected on a Rigaku MiniFlex 600 powder diffractometer (Rigaku Innovative Technologies, Inc., Auburn Hills, MI, USA) using a Cu-Kα source (λ = 1.5418 Å) over the 2θ range of 5° to 40° with a scan rate of 1°/minute. Contact angle measurements were performed on a Kruss Contact Angle Measuring System G2 (Krüss GmbH, Hamburg, Germany). The solid surfaces used for these measurements were dry-pressed Per-TAPB-PPI pellets (at 300 MPa for 10 s), and either water or dimethyl carbonate were used as liquids.

### 2.4. Electrochemical Study

#### 2.4.1. Electrodes Prepared by the Traditional Mixing Method

The active material (**Per-TAPB-PPI**, 0.5 mg), acetylene black (2 mg) and PVDF binder were ground together using a small amount of NMP, and the resulting paste was loaded on a copper foil sheet using a doctor blade technique. PVDF act as an effective dispersion agent to connect the electrode species together, ensure the adhesion of materials to the current collectors and provide mechanical integrity for the electrode. The electrodes were dried under vacuum at 50 °C for around 24 h and pressed using a mechanical press with a pressure of 75 bar (Metaserv Automatic mounting press, Model C190, Surrey, England).

#### 2.4.2. Electrodes Prepared by the Layer-by-Layer Method

Acetylene black (2 mg) and PVDF binder (0.1 mg) were ground using a small amount of NMP, and the resulting paste was loaded on a copper foil sheet using a doctor blade technique. Afterwards, the coated sheets were dried in a vacuum oven at 50 °C for 2 h to evaporate the excess NMP solvent. The active material (**Per-TAPB-PPI**) and PVDF binder (1%) were sonicated with NMP to form a homogenous slurry and dropwise added to the acetylene black/PVDF layer. The active material loading was ~0.28 mg·cm^−2^. The electrodes were dried under vacuum at 50 °C for around 24 h and pressed using a mechanical press (pressure, 75 bar).

#### 2.4.3. SC Fabrication

The symmetrical supercapacitors (SCs) were fabricated by sandwiching two identical working electrodes in a 2032 stainless steel coin cell in both scenarios. The electrolyte was 1 M LiPF6 in ethylene carbonate/ethyl methyl carbonate (EC/EMC) mixture (1:1, *v*/*v*) and a glass microfiber filter paper (GF/B, pore size: 1.0 μm) applied as the separator (see Figure 1). All the coin cells were assembled in a dry glove box filled with argon. The supercapacitor performances of as-built coin cells were evaluated on an Iviumstat Electrochemical Interface. Cyclic voltammetry (CV) and galvanostatic charge–discharge (GCD) assessments were applied within the potential range of 0.02–2.22 V and 0.2–1.8 V for the layered SC and traditional mixing SC, respectively. The electrochemical impedance spectroscopy (EIS) analysis was carried out by employing an amplitude of 5 mV over the frequency range from 0.1 Hz to 400 kHz. The cycling stability of the SCs were evaluated over 5000 cycles.

## 3. Results and Discussion

### 3.1. Porous Polymer Characterisation

Imide-bond formation after the polymerisation was confirmed with FT-IR spectroscopy. The characteristic vibrations of the amine-containing monomer TAPB and the anhydride containing monomer PTCDA were not observed in the resulting polymer, while characteristic imide-bond vibrations appeared (see Appendix A, Supporting Information for detailed information). In addition, **Per-TAPB-PPI** showed excellent stability when exposed to heat, measured by TGA (Appendix A). A temperature of 546 °C was observed at 5% weight loss for Per-TAPB-PPI, which is on par with the current state-of-the-art porous polyimides [24,34], and it emphasises the advantage of using imide-linkages in porous polymers. Finally, while the specific use of the monomer TAPB was mainly to promote porous framework formation, it may have added benefits to the final material in terms of optoelectronic and mechanical properties [35].

Nitrogen sorption experiments were conducted to assess the porosity of the synthesised polymer (Figure 2a). The adsorption isotherm showed a steep increase in N_2_ uptake at the low relative pressure region (<0.05 P/P_0_), indicating the presence of micropores. Additionally, **Per-TAPB-PPI** exhibited a further (>0.05 P/P_0_) slight increase in N_2_ uptake, which indicates the presence of larger (meso)pores. Pore size distribution (PSD) was calculated by fitting the nitrogen adsorption isotherm to a DFT model (Figure 2b). Specifically, the quenched solid DFT (QSDFT) carbon model was chosen for the PSD calculations since it takes surface roughness into account [36]. Considering that this polymer is largely amorphous (discussed later), the pore geometries are expected to vary, meaning that slit-like, cylindrical, and spherical pores all were considered in the model. Next to the significant presence of micropores within the PSD of **Per-TAPB-PPI**, a distribution around the maximum population corresponding to a pore size of 2.5 nm was observed. It should be noted that such PSDs are still estimates based on models that are initially developed for carbon materials. The calculated surface area according to the Brunauer–Emmett–Teller (BET) theory for **Per-TAPB-PPI** was 411 m^2^·g^−1^ (Appendix A).

While the above results approximate a quantitative description of the nanoporous polymer, we conducted scanning electron microscopy (SEM) measurements to obtain a better qualitative picture of the microscopic porous morphology (Figure 2c). **Per-TAPB-PPI** exhibits large porous aggregate structures originating from interconnected plate-like and spindle-like substructures. In addition to the micro- and mesoporous structures discovered by nitrogen sorption studies described earlier, these SEM images also reveal the presence of larger macropores. This hierarchical porous structure (micro-, meso- and macroporosity) promises effective utilisation of the polymer’s redox-active sites. The architecture of the polymer was further investigated by the use of powder X-ray diffraction (PXRD, Appendix A). The polymer exhibits a largely amorphous nature with no distinct crystalline domains, while increased diffuse intensity in the lower-angle regions of the PXRD pattern may indicate a short-ranged order of distances similar to the mesopore sizes observed in Figure 2b. Furthermore, the broad signal of **Per-TAPB-PPI** around 24° (corresponding to a distance of 3.7 Å) resembles the interlayer stacking signal often seen in covalent organic frameworks [37], which may indicate a sheet-like character. Although **Per-TAPB-PPI** does not show long-range crystalline order, we do expect that, based on these results and similar patterns observed in the literature, the polymer structure exhibits local stacked regions of neighbouring planes. The porous polymer architecture may influence the wettability of the material, which was investigated next. 

The electrolyte wettability of the **Per-TAPB-PPI** electrode is important for the electrochemical reactions at the electrolyte–electrode interface. We measured this property by performing swelling experiments through a contact angle measurement setup. Dry-pressed polymer pellets were used as solid surfaces. Unfortunately, a rough final surface is inherently linked to this sample preparation, which makes it difficult to obtain a defined contact angle from these measurements. However, the qualitative results that were obtained when comparing water or dimethyl carbonate on a **Per-TAPB-PPI** surface (Appendix A) exemplified the advantage of using carbonate esters as the electrolyte in the case of electrochemical applications. The referred figure shows the remarkable swelling behaviour of **Per-TAPB-PPI** when exposed to a droplet of dimethyl carbonate. On the other hand, water does not interact with the polymer in such a way and merely wets the polymer as a result of surface roughness. The hydrophobicity (extended aromatic structures) of the material is expected to be the main reason for this, which is strengthened by other research on vapour adsorption experiments of porous polyimides that show poor water uptake (compared to organic solvents such as benzene and hexane, with benzene/H_2_O selectivity values of 13–28 reported by Li and Wang) [38,39]. Thus, these experiments not only illustrate a good wetting of the polymers with the organic electrolyte but even a pore-opening behaviour and consequently increased access to the polymer’s active sites.

### 3.2. Supercapacitor Performance

The extensively characterised porous polymer **Per-TAPB-PPI** of the previous section has been implemented into two differently prepared electrodes in SCs: via a traditional mixing method and also via the layer-by-layer method. As indicated in the introduction, we expected significant compositional and morphological differences at the electrode’s surface that is exposed to the electrolyte. In order to confirm this, SEM micrographs were taken of the surfaces of the two different electrodes (Appendix A). The conventional electrode surface was comparatively smooth and displayed a slightly porous structure that originates from the conductive carbon network. In contrast, the layered electrode revealed the presence of embedded aggregated porous polymer particles over the surface. To verify that this assumption, energy dispersive X-ray spectroscopy (EDX) was used to assess both different electrode surfaces (Appendix A). The layer-by-layer-prepared electrodes contained a significantly higher amount of oxygen atoms at the surface, originating from the carbonyl groups of **Per-TAPB-PPI**. The larger availability of these redox-active carbonyl-groups is expected to improve the electrochemical performance of the layer-by-layer-prepared SCs when compared to the SC comprised of traditionally prepared electrodes.

The electrochemical performances of the SCs were studied using LiPF6 salt dissolved in a mixture of EC/EMC solvents as the electrolyte. Figure 3a,b show the CV curves for the **Per-TAPB-PPI**-based SCs prepared by the traditional mixing method and the layer-by-layer method, respectively, both at scan rates ranging from 10 to 200 mV·s^−1^. The CV curves of the SC fabricated from the traditional method show an electrical double layer capacitance (EDLC) with a minute pseudocapacitive contribution, which is apparent from the absence of clear redox peaks. In contrast, the layer-by-layer prepared **Per-TAPB-PPI** SC (Figure 3b) delivered EDLC, as well as a clear pseudocapacitance along with distinctive reversible redox humps at ~0.95 V for cathodic and ~1.20 and ~1.60 V for anodic peaks. First, the EDLC of both SCs can be attributed to the large surface area and nanoporous architecture of **Per-TAPB-PPI**, allowing the diffusion and adsorption of a high number of Li^+^ ions. In addition, the contribution of acetylene black could be recognised in both scenarios in terms of EDLC, with a more prominent involvement expected in the traditional SC. Additionally, the presence of a sizeable π-conjugated system could further facilitate the electrical conductivity and hence increase the Li^+^ ion adsorption to the active material [40,41]. Secondly, the pronounced pseudocapacitive contribution in the layered SC CVs indicates surface faradic redox reactions of Li^+^ ions with carbonyl functionalities of the perylene diimide-based polymers (Figure 2). This also explains the overall higher current compared to the traditional **Per-TAPB-PPI** SC. Furthermore, the minor faradic contribution in the conventional SC is likely caused by the hindrance of the acetylene black/PVDF matrix, limiting the availability of the active sites of the **Per-TAPB-PPI** polymer.

To further investigate the electrochemical performance of the **Per-TAPB-PPI** SCs, GCD measurements were carried out at various current densities (Figure 3c,d). For the traditional SC, a nearly perfect triangular-shaped profile was observed. Such a curved shape indicates a pure EDLC character, which reinforces the CV results. On the other hand, the GCD curves of the layered electrode displayed prominent nonlinear characteristics, confirming the pseudocapacitive contribution. Moreover, the layer-by-layer prepared SC exhibited longer discharge times at comparatively higher current densities than those of the traditional SC. Additionally, the layered SC’s discharge curves show a rapid voltage decay at the early stage due to non-faradic electrostatic discharge (i.e., adsorption and diffusion processes) and a final slow potential drop due to the faradic discharge [42,43]. From these GCD plots, the specific capacitance C_s_ of both SCs was estimated using Equation (1) [1].
(1)$$\mathrm{Cs}=4\frac{I}{m\left(dV/dt\right)}$$
where *I* (A) is the constant discharge current, *t* (s) is the discharge time, *V* (V) is the potential window, and *m* (g) is the active material mass in both electrodes. Furthermore, the energy (E) and the power (P) densities for the SC cells were estimated using E = 0.125 C_s_ (ΔV)^2^/3.6 and P = 3600 E/Δt, respectively [44,45].

As presented in Figure 3e, the highest specific capacitance C_s_ of the traditional **Per-TAPB-PPI**-based SC was 178 F·g^−1^ at a current density of 0.1 A·g^−1^ while the layered SC exhibited an outstanding capacitance of 388 F g^−1^ at a specific current density of 0.4 A·g^−1^. The resulting higher C_s_ of the layered SC follows directly from the observations in the CV and GCD plots, with the most prominent reason being the additional pseudocapacitive behaviour that this SC shows. The significant differences in C_s_ values between the differently prepared electrodes exemplify the electrode fabrication method greatly influences its electrochemical performance. In our case, **Per-TAPB-PPI** benefits from being in direct contact with the electrolyte (rather than being obstructed in the acetylene black/PVDF matrix), since the fast faradic surface reactions are then better exploited. Moreover, the energy (E) and power (P) densities for the SC devices were evaluated and plotted in Figure 3f. As presented, the layer-by-layer prepared **Per-TAPB-PPI** SC device exhibited an excellent maximum energy density of 65 Wh·kg^−1^ at 0.4 A·g^−1^ and a high power density of 2.2 kW·kg^−1^ at 1.6 A·g^−1^. Similarly, the maximum energy density for the traditionally prepared **Per-TAPB-PPI**- SC device was only 16 Wh·kg^−1^ at a current density of 0.1 A·g^−1^ along with a power density of 2.6 kW·kg^−1^ at 1.6 A·g^−1^. According to the results, we demonstrated a substantial energy density improvement in the layer-by-layer prepared **Per-TAPB-PPI** SC device, confirming the enhanced electrochemical performance. 

The cyclability test of the electrode prepared by the traditional and by the layer-by-layer methods are illustrated in Figure 4. Mass loadings of the electrodes were optimised for a balanced capacitance and cycle stability. During the cyclic test, the layered SC was first run at a current density of 0.8 A·g^−1^ over 5 cycles to activate the SC and a remarkable 90.6% capacitance retention was observed after 5000 cycles at a current density of 1.6 A·g^−1^. Similarly, the traditional SC was activated at 0.4 A·g^−1^ over 5 cycles, and the 93.9% capacitance retention was obtained after 5000 cycles at a current density of 0.8 A·g^−1^. The layered SC demonstrates the excellent electrochemical performance during cycling at a higher current density compared to the traditional electrode, which displays fast electrochemical kinetics that arise from the direct contact between the electrolyte and the porous polymer layer. The calculated cyclability of **Per-TAPB-PPI** are comparable with other COF-based electrodes prepared without any hybridisation with carbon materials as illustrated in Appendix A (supporting information), indicating the feasibility of our new materials and electrode fabrication method on improving the capacitance without sacrificing the cyclability of the electrode. A detailed post-mortem analysis of the electrode surface after cycling experiments, as has been studied by the group of Yoo [46], is able to provide additional insights in the effect of induced stress and strain to porous polymer materials. Our group is currently investigating this and aims to report this in future work.

A final characterisation method we used to elucidate the electrochemical performance of the differently prepared electrodes was electrochemical impedance spectroscopy (EIS), since it provides information on the equilibrated states of the SCs. The experiments were performed over a frequency range of 0.01–400 kHz, and the resulting impedance plots (Nyquist plots) are presented in Figure 5a,b. Overall, the near-vertical lines for the lower frequency range (Figure 5a) of both SCs indicate capacitive behaviour and efficient ion diffusion between the electrolyte and the electrode surface [47,48,49,50,51]. Furthermore, the higher frequencies in the Nyquist plots (Figure 5b) provide information about the serial (R_S_) and charge transfer (R_CT_) resistances in the electrodes. Here, the intersect with the Z’ axis gives R_S_, which originates from the internal resistance of the electrode materials. Both electrodes show relatively low internal resistances: 4.6 Ω for the layered and 6.2 Ω for the traditional electrode. These results suggest the presence of efficient conductive pathways between the polymer (π electrons) and conductive carbon. In addition, a slightly higher R_CT_ was observed for the layered electrode (3.0 Ω) compared to that of the traditional electrode (0.6 Ω). The lower charge transfer resistance in the traditional electrode is likely an effect of the large contribution by the highly conductive carbon matrix. Furthermore, the R_CT_ of both the layered and traditional **Per-TAPB-PPI** SCs were significantly lower than that of previously reported organic polymer electrodes including a hydroquinone-based COF TpPa-(OH) (37.5 Ω) [52], and a benzimidazole-based COF TpDAB (24.8 Ω) [53]. This contrast is likely an effect of molecular structure, rather than polymer architecture or device preparation procedure, since perylene diimide derivatives are among the most common acceptor units because of their high electron affinity and electron mobility. Finally, additional experiments on a range of related polymers with perylene diimide units are required to link the promising performance of this polymer to its structure.

## 4. Conclusions

In this research, we investigated the effects of a novel layer-by-layer electrode preparation method on the performance of a porous polymer-based supercapacitor. By physically separating the redox-active porous polymer layer and the electron-conducting carbon layer, we created better accessible pathways for ions to diffuse towards the electrodes. As a result, faradic surface reactions were contributing significantly to the overall supercapacitor performance. In addition, by exposing the polymer directly to the electrolyte, the advantages of a porous polyimide polymer (i.e., nanoporosity and redox-activity) were much more exploited. In planned future research, we aim to use a variety of porous polymers in layer-by-layer prepared electrodes to elucidate the correlation between polymer architecture and electrochemical performance. We expect that this electrode preparation method allows the observation of more apparent effects originating from the polymer itself, which bridges the gap between molecular structure and electrochemical performance. 

## Data Availability

Data is contained within the article.

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
