# Peer review of "Layer-by-Layer Electrode Fabrication for Improved Performance of Porous Polyimide-Based Supercapacitors"

_materials, 2021, doi:10.3390/ma15010004_

Round 1

Reviewer 1 Report

In this article, the authors demonstrate a novel layer-by-layer manufacturing strategy that allows direct contact between electrolyte and the active material by coating a conductive layer and an active material layer on the copper foil. Benefit from the high specific surface area of 411 m2/g and the porous structure, the obtained electrode displays good electrochemical performance. This method is interesting and the manuscript is well organized. Therefore, I suggest that the manuscript could be published before a minor revision:

  1. Why using porous perylene diimide-based polyimide (Per-TAPB-PPI) as the electrode material, any particular reasons? The authors should give more explanation of Per-TAPB-PPI in the Introduction part.
  2. A High-resolution SEM image should be provided to illustrate the porosity of the material (Figure 2c).
  3. The mass loading of the electrode was about 0.28 mg cm-2. Usually, commercial electrodes need a high mass loading, such as > 5 mg cm-2 [ Mater. Chem. A., 2020, 8, 1176-1183]. Therefore, I suggest that the authors provide the specific capacitance of a high mass loading electrode.
  4. Why did the authors compare the cycling stability of the two electrodes using different current densities? The cycling tests of the layered and traditional Per-TAPB-PPI SCs should be performed at the same current density (Figure 3e).

Reviewer 2 Report

The manuscript is well written, and the data looks good. But before publish on this journal, it should address the following comments.

  1. Why PVDF is used as the binder? what effect will PVDF have on the performance of the electrode?
  2. The TEM test should be added to clearly show the uniformity and pore size of the micropores.
  3. Please explain in detail why XRD shows that per-TAPB-PPI has a sheet-like character.
  4. The water absorption test should be supplemented to prove that porous polyimide has low water absorption.
  5. What are the advantages of porous polymer particles embedded in the layered electrode.
  6. Please refer to the newly paper about TAPB-PI "Construction of micro-branched crosslink fluorinated polyimide with ultra-low dielectric permittivity and enhanced mechanical properties, Express Polymer Letters, DOI: 10.3144/expresspolymlett.2022.12"

Reviewer 3 Report

In this manuscript, Abdelkader and coworkers reported a layer-by-layer fabrication technique to prepare a kind of "layered" electrodes for supercapacitor. Through this approach, direct contact with the electrolyte and polymer material is greatly enhanced, which allowed a significant contribution of fast faradic surface reactions to the overall capacitance. The specific capacitance of 388 F g-1 was achieved at a current density of 0.4 A g-1. However, some detailed characterizations are not full filled. Overall, this work is well organized. The manuscript is recommended to be published after major revisions.

Detail comments can be found as below,

  1. The structure of Per-TAPB-PPI should be characterized by solid-state 13C NMR.
  2. The quality of SEM image is not good enough, the sheet-like morphology (page 7, line 202) of Per-TAPB-PPI is not clear.
  3. The porous character of Per-TAPB-PPI also needs to be confirmed by TEM.
  4. In the preparation of electrodes materials, the amount of conductive carbon in "layered" electrodes was higher than that of traditional mixing method. Owing to the some EDLC capacitance of acetylene black, thus, the comparison of two different methods is not meaningless.
  5. Lack of energy density and power density performance diagram.
  6. Figure S5 is missing the measurement of contact angle degrees, and 217th line "compared to organic solvents such as benzene and hexane" does not give data support.

Reviewer 4 Report

The manuscript title: “Layer by layer electrode fabrication for improved performance of porous polyimide-based supercapacitors” by Niranjala et al. did a detailed study on electrode fabrication methods for supercapacitors. The present manuscript is fascinating. The author’s presentation and result, and discussion part are impressive. However, authors should address the following minor issues in the manuscript before publishing it in the Journal of materials.”

  1. Many improper signs and notations were present, and some typo errors should also be corrected before publication. (ex. use xrd degree symbols on every point of results.)
  2. Authors must produce high magnification morphology presentations.
  3. In the contact angle study, the performances were not at their best level authors should reconsider the retest of the study. (10.3390/polym12091871)
  4. In the introduction, the authors could add more details about layer-by-layer coating advantages in supercapacitors (10.1038/s41598-019-41203-3).
  5. It Is possible then authors can produce post morphology studies (10.1016/j.apcatb.2021.120405).

Round 2

Reviewer 3 Report

Abdelkader and coworkers submitted the good manuscript with appropriate results, and answer these comments carefully. I am highly satisfied with this revised version. I recommended to accept this manuscript.